



# Stochastic analysis of cone penetration tests in snow

Pyei Phyo Lin[1], Isabel Peinke[2], Pascal Hagenmuller[2], Matthias Wächter[1], M. Reza Rahimi Tabar[1,3], and Joachim Peinke[1]

[1]ForWind, Institute of Physics, University of Oldenburg, Oldenburg, Germany
[2]Univ. Grenoble Alpes, Université de Toulouse, Météo-France, CNRS, CNRM, Centre d'Études de la Neige, Grenoble, France
[3]Department of Physics, Sharif University of Technology, Tehran 11155-9161, Iran

**Correspondence:** Pyei Phyo Lin (pyei.phyo.lin@uol.de)

**Abstract.** Cone penetration tests have long been used to characterize the snowpack stratigraphy. With the development of sophisticated digital penetrometers such as the Snow MicroPenetrometer, vertical profiles of snow hardness can now be measured at a spatial resolution of a few microns. At this high vertical resolution and by using small penetrometer tips, more and more details of the penetration process get resolved, leading to much more stochastic signals. An accurate interpretation of these

signals regarding snow characteristics requires employing advanced data analysis. Here, the failure of ice connections and the pushing aside of separated snow grains during cone penetration lead to a combination of a) diffusive noise, as in Brownian motion, and b) jumpy noise, as proposed by previous dedicated inversion methods. The determination of the Kramers-Moyal coefficients allows differentiating between diffusive and jumpy behaviors and determining the functional resistance dependencies of these stochastic contributions. We show how different snow types can be characterized by this combination of

highly-resolved measurements and data analysis methods. In particular, we show that denser snow structures exhibited a more collective diffusive behavior supposedly related to the pushing aside of separated snow grains. On lighter structures with larger pore space, the measured hardness profile appeared to be characterized by stronger jump noise probably related to breaking single cohesive bonds. The proposed methodology provides new insights into the characterization of the snowpack stratigraphy with cone penetration tests.

## 1  Introduction

Snow is an essential component of our environment and can significantly impact our life: from the wishful dream of white Christmas to the misfortune of avalanche accidents. Having a closer look at snow, one discovers many microstructural patterns and realizes that snow on the ground undergoes constant evolution (Colbeck et al., 1990). The snow microstructure fully controls its physical and mechanical properties, which are essential for diverse applications, such as avalanche forecasting

(Schweizer et al., 2003). A snowpack is typically structured in numerous layers composed of different snow types, where such stratigraphy will determine the snowpack stability. Cone penetrations tests have long been used to characterize the snowpack stratigraphy (Bader and Niggli, 1939). The Snow Micro Penetrometer (SMP) can perform cone penetration tests of snow in the field (Schneebeli et al., 1999). It measures the force needed to drive a cylinder with a millimetric conic tip into the snowpack.





With its high resolution (250 measurements per mm), the measured force or hardness is supposed to be linked to the snow
microstructure (Johnson and Schneebeli, 1999). A typical consequence of such a high-precision measurement is that more and
more details of the penetration process get resolved, leading to much more stochastic signals. In this context, it is of particular
interest to employ advanced data analysis to find out how different kinds of stochastic signals are related to different snow
types.

Johnson and Schneebeli (1999) developed the first model to estimate micromechanical properties of snow from measured
penetration profiles. They assumed that the material compaction is negligible and that the penetration resistance is only com-
posed of friction between the penetrometer and snow grains and a superposition of spatially uncorrelated and identical brittle
failures of individual snow microstructural elements (e.g., the bonds between the snow grains). Marshall and Johnson (2009)
extended the theory of Johnson and Schneebeli (1999) to account for simultaneous ruptures by Monte-Carlo simulations. They
precisely resolved the snow micro-mechanical parameters, such as the deflection length, rupture force, and rupture intensity.
Löwe and van Herwijnen (2012) re-stated the pioneering idea of Johnson and Schneebeli (1999) and described the fluctuating
penetration hardness as a Poisson shot noise process. In their model, the micromechanical parameters can be simply derived
from the cumulants and the co-variance of the penetration signal. Peinke et al. (2019) further extended the homogeneous Pois-
son process of Löwe and van Herwijnen (2012) so that the scale of variation of the rupture intensity can be decoupled from the
scale of variations of the deflection length and the rupture force. The developments of these models had important applications
in snow science. Indeed, Proksch et al. (2015) related the micro-mechanical parameters derived from the SMP measurements
to ones of the most critical snow characteristics, namely density, specific surface area, and structural correlation length. These
relations are now routinely used to quantify the snowpack stratigraphy (e.g., Calonne et al., 2020). Besides, Reuter et al. (2019)
estimated elastic modulus and fracture energy from the micro-mechanical parameters, which can then be used to compute point
snow stability for avalanche hazard assessment (e.g., Reuter et al., 2015).

Here, we consider the measured fluctuating hardness as a consequence of summing up the interactions between the pen-
etrometer tip and individual snow particles. We describe this penetration process in analogy to the well known Brownian
motion (Einstein, 1905), where a microscopically visible particle suspended in the fluid is moving randomly due to the sum of
several collisions with the molecules in the fluid, as illustrated in Figure 1. Like for the Brownian motion, the sum of distinct
events causes a stochastic process with a trend and some additional noise related to micro-events. Such a stochastic process
is driven by white noise and is known as a Langevin process (see Gardiner (1985) and Sect. 2). While very similar elemen-
tary collision events are assumed for classical Brownian motion, we also need to consider brittle failures of individual snow
microstructural elements (bonds between snow grains, crushing of grain clusters), which cause sharp declines in penetration
hardness. These brittle failures were modeled as a Poisson shot noise process by Löwe and van Herwijnen (2012); Peinke et al.
(2019). Jump noise acts as discontinuous paths inside the diffusion process, and low jump events can be considered as Pois-
son distributed noise. The idea of this work is to employ a method that allows estimating the underlying stochastic differential
equation from empirical data and differentiating between a Langevin (pure diffusive) or a jump-diffusion process (Anvari et al.,
2016). With such an advanced analysis, we aim at a more detailed snow characterization from cone penetration tests[1].

---

[1]A direct comparison of our stochastic approach with the works based on shot noise is out of the scope of this paper.





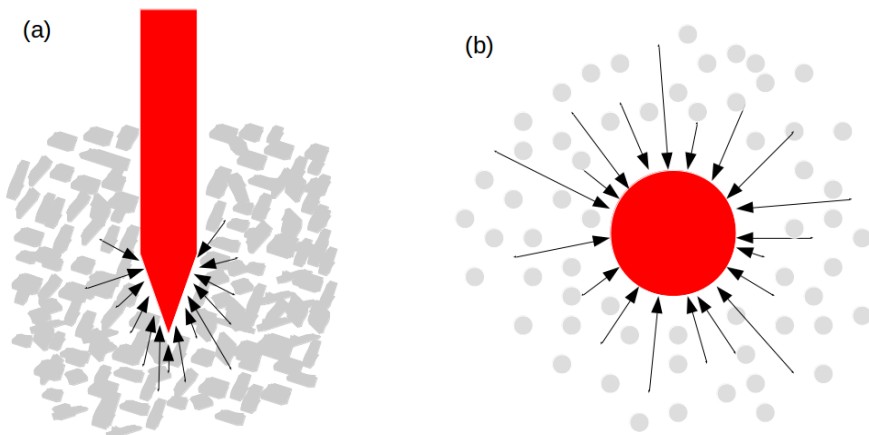

**Figure 1.** (a) Penetration resistance caused by the interactions of the snow particles (gray) with the penetrometer tip (red); (b) Brownian motion where a microscopically visible particle (red) suspended in the fluid is moving randomly due to the collisions with the molecules (gray) in the fluid

The article is organized as follows. In Sect. 2, we summarize the stochastic analysis method and show how it is possible to distinguish between diffusive and jump noise. In the Sect. 3, the method is applied to centimetric snow samples whose microstructure is also captured by tomography. In Sect. 4, as a proof of concept, the SMP profile of a natural snowpack is analysed with this technique.

## 2  Stochastic Method

A stochastic process $x(t)$ can be described through stochastic differential equations. This section explains the equation used to model cone penetration tests in snow. Since the SMP is driven by a motor with constant speed $u = \frac{dz}{dt}$ ($z$ is depth, $t$ is time) and samples the measurement every 4 $\mu$m, the measured penetration force or snow hardness $R$ is considered as the depth dynamics $R(z(t))$ and handled like a stochastic variable $x(t)$.

### 2.1  Langevin Equation

A diffusive process $x(t)$, which is a continuous stochastic process, follows the Langevin equation, where for a small step size $dt$ has the following expression (Risken, 1984; Friedrich et al., 2011; Tabar, 2019):

$$\mathrm{d}x(t) = D^{(1)}(x,t)\,\mathrm{d}t + \sqrt{D^{(2)}(x,t)}\,\mathrm{d}W_t, \qquad (1)$$

where $D^{(1)}(x,t)$ and $D^{(2)}(x,t)$ are the drift and the diffusion coefficients, respectively, and $W_t$ is the Wiener process. The drift term $D^{(1)}$ describes how fluctuations relax to the local mean values of $x$, defined by $D^{(1)}(x,t) = 0$. The diffusion term $D^{(2)}$ represents the amplitude of the noise. The coefficients $D^{(1)}$ and $D^{(2)}$ are also known as the first and second order Kramers-





Moyal (KM) coefficients, respectively. In general, KM coefficients can be directly determined from the given data $x(t)$ using
their definitions as of the conditional incremental average (Friedrich et al., 2011; Tabar, 2019), i.e.,

$$K^{(j)}(x,t) = \lim_{\Delta t \to 0} \frac{\left\langle (x(t+\Delta t) - x(t))^j |_{x(t)=x} \right\rangle}{\Delta t}. \tag{2}$$

Further details on methods of this estimation can be found in Friedrich et al. (2011); Rinn et al. (2016)[2]. The Langevin equation
describes a continuous diffusion process where $K^{(j)}(x,t) = 0$ for $j \geq 3$ and $D^{(j)}(x,t) = K^{(j)}(x,t)$. According to Pawula's
theorem, all KM coefficients $K^{(j)}$ are vanishing for $j \geq 3$ if $K^{(4)}(x,t) = 0$ (Risken, 1984; Pawula, 1967). In our case, however,
the higher order coefficients are not always vanishing, hence we extend the discussion to the jump-diffusion process (see Sect.
2.2).

For the considered SMP data, the drift term $D^{(1)}$ in Eq. 1 describes how hardness fluctuations relax to the local mean values
of hardness $R$, defined by $D^{(1)}(R,z) = 0$. The diffusion $D^{(2)}$ term represents the amplitude of the hardness fluctuations. The
coefficients $D^{(1)}$ and $D^{(2)}$ are $z$-dependent for non-stationary (inhomogeneous) processes. Here, we assume that for a chosen
small depth ($z$) interval $D^{(1)}(R,z)$ and $D^{(2)}(R,z)$ only depend on $R$ (similarly to Löwe and van Herwijnen (2012) for their
shot noise model).

## 2.2 Jump-Diffusion Dynamics

Typically, when the signal of a stochastic process presents sharp changes at some instant (discontinuities events), higher order
Kramer-Moyal coefficients (especially $K^{(4)}(x,t)$) become non-negligible. In this case, an extension to the Langevin-type
modeling with the additional jump noise is needed, (see Tankov, 2003; Stanton, 1997; Johannes, 2004; Bandi and Nguyen,
2003; Anvari et al., 2016; Tabar, 2019). Such a jump-diffusion dynamics is given by the following stochastic differential
equation:

$$dx(t) = D^{(1)}(x,t)\,dt + \sqrt{D^{(2)}(x,t)}\,dW_t + \xi\,dJ_t, \tag{3}$$

where, again, $D^{(1)}$ and $D^{(2)}$ are the drift and the diffusion coefficients, respectively, and $W_t$ is the Wiener process. The quantity
$\xi$ is the size of the jump noise and is assumed to be normally distributed, i.e., $\xi \sim N(0, \sigma_\xi^2)$ where $\sigma_\xi^2(x,t)$ is the so-called jump
amplitude. The term $J_t$ is the Poisson jump process, which is the zero-one jump process with a jump rate (or intensity) $\lambda(x,t)$
(Hanson, 2007; Tabar, 2019).

For jump-diffusion processes, the drift and diffusion coefficients ($D^{(1)}, D^{(2)}$), the jump rate $\sigma_\xi^2$ and amplitude $\lambda$ are related
to the KM coefficients as (Anvari et al., 2016):

$$D^{(1)}(x,t) = K^{(1)}(x,t), \tag{4}$$

---

[2]KM coefficients for Langevin equation are defined as $K^{(j)}(x,t) = D^{(j)}(x,t) = \frac{1}{j!}\lim_{\Delta t \to 0} \frac{\left\langle (x(t+\Delta t) - x(t))^n |_{x(t)=x} \right\rangle}{\Delta t}$ in Friedrich et al. (2011); Rinn
et al. (2016). In order to make it consistent with the jump-diffusion process, our definition is differed by a factor of $\frac{1}{j!}$, in which $dW_t = \Gamma(t)\cdot dt$ where $\langle\Gamma(t)\rangle =$
0 and $\langle\Gamma(t)\Gamma(t')\rangle = \delta(t-t')$. The corresponding Fokker-Planck equation will be $\frac{\partial}{\partial t}p(x,t) = -\frac{\partial}{\partial x}\left[D^{(1)}(x,t)\,p(x,t)\right] + \frac{1}{2}\frac{\partial^2}{\partial x^2}\left[D^{(2)}(x,t)\,p(x,t)\right]$.



$$D^{(2)}(x,t) + \lambda(x,t)\langle\xi^2\rangle = K^{(2)}(x,t), \tag{5}$$

$$\lambda(x,t)\langle\xi^j\rangle = K^{(j)}(x,t), \quad \text{for } j > 2. \tag{6}$$

The estimate of the drift coefficient is the same for the diffusion process (Eq. 1) and the jump-diffusion process (Eq. 4). Jump amplitude $\sigma_\xi^2$ and rate $\lambda$ can be estimated by using Eq. 6 with $j = 4$ and $j = 6$, and the Wick's theorem for Gaussian random variables which states that $\langle\xi^{2n}\rangle = \frac{(2n)!}{2^n n!}\langle\xi^2\rangle^n$:

$$\sigma_\xi^2(x,t) = \frac{K^{(6)}(x,t)}{5K^{(4)}(x,t)}, \tag{7}$$

$$\lambda(x,t) = \frac{K^{(4)}(x,t)}{3\sigma_\xi^4(x,t)}. \tag{8}$$

To improve the estimation of KM coefficients $K^{(j)}(x,t)$ and in particular of high-order coefficients, the Nadaraya-Watson estimator, which is a kernel estimator, can be used (Nadaraya, 1964; Watson, 1964):

$$K^{(j)}(x,t) = \lim_{\Delta t \to 0} \frac{\sum_i k(\frac{x_{i\Delta t}-x}{h})(x_{(i+1)\Delta t} - x_{i\Delta t})^j}{\sum_i k(\frac{x_{i\Delta t}-x}{h})\Delta t}, \tag{9}$$

where we use a Gaussian kernel $k(u)$ here (Tabar, 2019). With the kernel-based method the conditional moments can be
calculated more smoothly by controlling the kernel bandwidth $h$ (Lamouroux and Lehnertz, 2009). In our analysis, we use the kernel bandwidth $h = 0.3$.

For the considered SMP measurements, we also assume that for a chosen small depth ($z$) interval $D^{(1)}(R,z)$, $D^{(2)}(R,z)$, $\sigma_\xi^2(R,z)$ and $\lambda(R,z)$ only depend on $R$. The stochastic differential equation for the jump diffusion process thus reads, on small depth intervals, as

$$\mathrm{d}R(z) = D^{(1)}(R)\,\mathrm{d}z + \sqrt{D^{(2)}(R)}\,\mathrm{d}W_z + \xi\,\mathrm{d}J_z, \tag{10}$$

with an analog interpretation of the drift and diffusion terms ($D^{(1)}$, $D^{(2)}$) as for the purely diffusive process (Eq. 1), but now extended by a jump noise term. The jump rate $\lambda$ has the dimension of $\frac{1}{[Z]}$ and can be related to the shot noise intensity described by Löwe and van Herwijnen (2012); Peinke et al. (2019). Typically, $\lambda dz$ corresponds to the stochastic average of the number of jumps for a penetration increment of depth $dz$ (Anvari et al., 2016; Tabar, 2019). The jump amplitude $\sigma_\xi^2$ represents the square
of the typical size of a jump. Note that the jump can be negative (failure of a microstructural element) or positive (loading of a microstructural element). Here, we do not consider any progressive loading of a microstructural element as described by Löwe and van Herwijnen (2012); Peinke et al. (2019) with the microstructural deflection length $\delta$. Here, the loading of a microstructural element and its contribution to penetration hardness is somehow instantaneous.



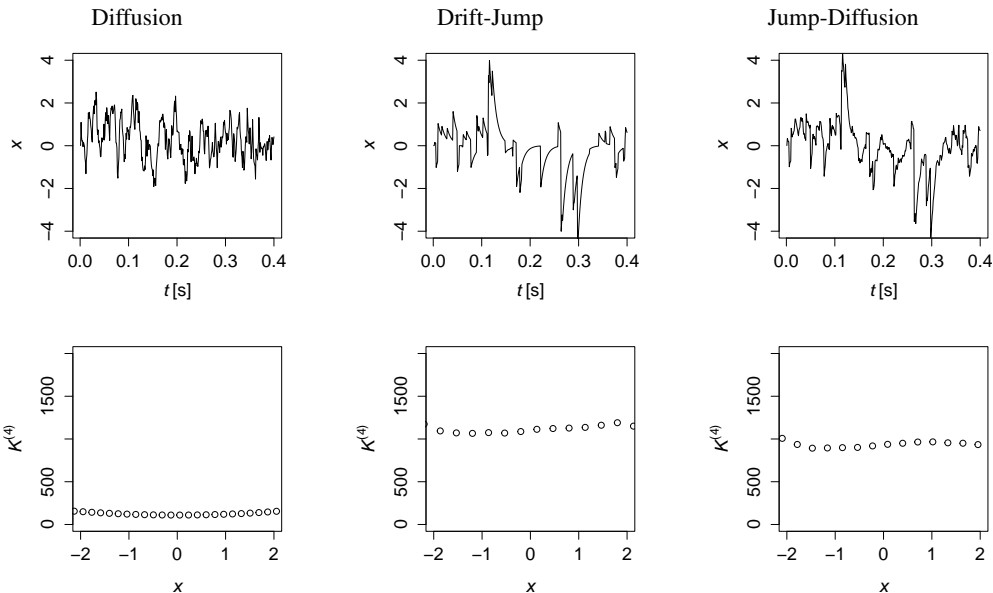

**Figure 2.** Normalized timeseries of OU processes with only diffusion, only jump (drift-jump) and jump-diffusion terms (top) and their fourth-order KM coefficients $K^{(4)}$ (bottom). $K^{(4)}$ of the diffusion process is negligible compared to jump and jump-diffusion processes. For all three examples, we use the same noises in the stochastic part of the SDE.

## 2.3 Synthetic examples

In this section, we illustrate how diffusive and jump noises affect the stochastic fluctuations on a synthetic example. An Ornstein-Uhlenbeck (OU) process $x$ is described by a stochastic differential equation (SDE) with linear relaxation dynamics and an additive uncorrelated noise. With an additional jump term, it is defined as

$$\mathrm{d}x = -\gamma x \, \mathrm{d}t + \sqrt{D} \, \mathrm{d}W_t + \xi \, \mathrm{d}J_t. \tag{11}$$

where $\gamma$ is the relaxation rate, $D$ the constant diffusion coefficient and $W_t$ a scalar Wiener process. As above, the noise
$\xi \sim N(0, \sigma_\xi^2)$ is assumed to be normally distributed with the constant variance or jump amplitude $\sigma_\xi^2$. $J_t \sim P(\lambda t)$ is the Poisson jump process, which is zero-one jump process with constant jump rate $\lambda$.

Three synthetic time series of the OU-jump-diffusion process are generated for $\Delta t = 10^{-3}\,\mathrm{s}$ with $\gamma = 100\,\mathrm{s}^{-1}$, $D = 10\,\mathrm{s}^{-1}$ and for the additional jump terms with $\lambda = 100\,\mathrm{s}^{-1}$ and $\sigma_\xi^2 = 1$. The generated data are normalized with their respective standard deviation. In Fig. 2 the normalized timeseries of the OU-jump-diffusion processes are shown. (Left a pure diffusion pro-
cess, middle a pure jump process and right, the combined jump-diffusion process is shown.) The fourth-order KM-coefficients $K^{(4)}(x)$ of each timeseries are also plotted in Fig. 2 bottom row. $K^{(4)}$ of the diffusion process is negligible small compared to drift-jump and jump-diffusion processes.

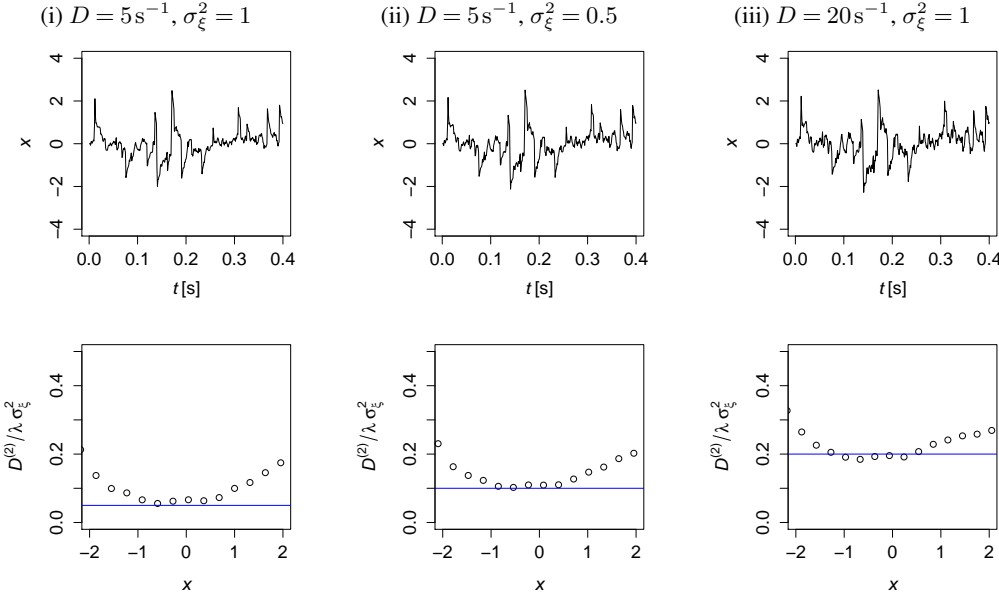

**Figure 3.** Normalized timeseries of OU-jump-diffusion processes with $\Delta t = 10^{-3}\,\mathrm{s}$, $\gamma = 100\,\mathrm{s}^{-1}$, $\lambda = 100\,\mathrm{s}^{-1}$, (i) $D = 5\,\mathrm{s}^{-1}$, $\sigma_\xi^2 = 1$, (ii) $D = 5\,\mathrm{s}^{-1}$, $\sigma_\xi^2 = 0.5$ and (iii) $D = 20\,\mathrm{s}^{-1}$, $\sigma_\xi^2 = 1$ (top) and the corresponding ratio of diffusion and jump noise $\frac{D^{(2)}}{\lambda \sigma_\xi^2} = \frac{D}{\lambda \sigma_\xi^2}$ (bottom). Dotes are results from the KM-coefficients, the blue line the theoretical values fiven by the constants. For all three examples, we use the same noises in the stochastic part of the SDE.

For a jump-diffusion process another parameter that we are going to consider is the ratio of diffusion and jump noise $\frac{D^{(2)}}{\lambda \sigma_\xi^2}$, which becomes here for our OU-jump-diffusion process $\frac{D}{\lambda \sigma_\xi^2}$. To proof evidence of our method based on the KM coefficients

of Eq. 5 to Eq. 8 three pairs of parameters (i) $D = 5\,\mathrm{s}^{-1}$, $\sigma_\xi^2 = 1$, (ii) $D = 5\,\mathrm{s}^{-1}$, $\sigma_\xi^2 = 0.5$ and (iii) $D = 20\,\mathrm{s}^{-1}$, $\sigma_\xi^2 = 1$ are chosen. The other parameters are the same as previous example. The normalized timeseries of these examples are plotted in Fig. 3 top row. In Fig. 3 bottom row the ratio of diffusion and jump noise $\frac{D}{\lambda \sigma_\xi^2}$ estimated from each timeseries are compared to the expected values (blue lines). As we used normalization and the same noises in simulation, all timeseries are very similar, however, one can observe clearly the more noisy fine structure in case (iii) where the diffusion coefficient is larger.

## 3  Application to snow measurements

In this section, our main aim is to show how the jump-diffusion model can be used to distinguish snow types from hardness profiles measured with the SMP. First, small snow samples whose microstructure is also fully characterized by tomography before being measured by the SMP, are used to test the developed methodology. Second, as a proof of concept, we analyse one penetration profile of a snowpack measured in the field and we provide the subsequent profile of microstructural parameters.

Last, the results are discussed.





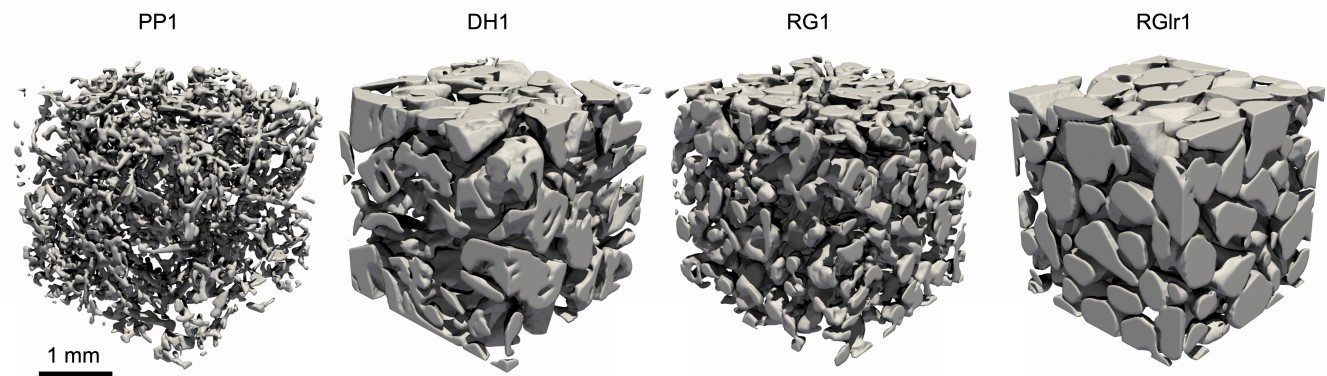

**Figure 4.** Three-dimensional view of the microstructure of some representative samples: Precipitation Particles (PP1), Depth Hoar (DH1), Rounded Grains (RG1) and large Rounded Grains (RGlr1). The ice matrix is shown in grey, the pore space is transparent. The shown sub-samples are cubic with a side-length of 3 mm. Details on the data acquisition can be found in (Peinke et al., 2020)

.

## 3.1 Laboratory samples

### 3.1.1 Measurement Data

We test several snow samples composed of four different natural snow types, namely Precipitation Particles (PP), Depth Hoar (DH), Rounded Grains (RG), and large Rounded Grains (RGlr). The samples are prepared by sieving snow into small sample holders (diameter and height of 20 mm) and letting them sinter for a couple of days. Their microstructure is captured with X-ray tomography at a nominal resolution of 15 $\mu$m (Fig. 4). The cone penetration test is conducted with a modified version of the SMP, as shown in Fig. 5. More information on the sample preparation and the SMP measurement can be found in the study of Peinke et al. (2020). The main sample properties are summarized in Tab. 1 and the measured profiles (one example for each snow type) are plotted in Fig. 6. The first 4 mm are affected by the progressive penetration of the conic tip and are not considered in the stochastic analysis (Peinke et al., 2019). The remaining profiles are divided into smaller segments of depth of 10 mm.

To work out the significance of advanced stochastic features for snow, we focus on the fluctuations of the penetration profiles. Each profile is first detrended. The trend $\bar{R}$ is computed as the convolution of the original signal with a Gaussian kernel with a standard deviation of 0.6 mm. The fluctuation amplitude $\sigma_R$ is computed as the standard deviation of $R - \bar{R}$. The detrended profiles are defined as $R' = \frac{R - \bar{R}}{\sigma_R}$. The detrended profiles $R'$ are characterized by zero mean and a standard deviation of 1. The average value of $\bar{R}$ on the segment and the value of the standard deviation $\sigma_R$ are shown for each segment in Tab. 1. The detrended profiles $R'$ are shown in Fig. 7, for all four snow types.

To estimate errors, we divide the detrended and normalized data into different sub-samples. Given are 2 PP, 3 DH, 3 RG and 6 RGlr measurement profiles. These profiles are separated into smaller segments, which finally gives 4 PP, 5 DH, 5 RG and 6



**Table 1.** Overview of the detailed properties of the used snow samples. Snow types are classified according to the international classification of snow on the ground Fierz et al. (2009). The density and specific surface area (SSA) are derived from the tomographic images Peinke et al. (2020). Additionally, the standard deviation $\sigma_R$ of the detrended profiles are calculated.

| Sample name | Snow type | Sieve size | Density | SSA | $\sigma_R$ |
|---|---|---|---|---|---|
| | | [mm] | [kg m$^{-3}$] | [m$^2$ kg$^{-1}$] | [kPa] |
| PP1 | Precipitation Particles | 1.6 | 92 | 53.5 | 0.55 |
| PP2 | Precipitation Particles | 1.6 | 137 | 41 | 0.81 |
| DH1 | Depth Hoar | 1.6 | 345 | 16.9 | 4.78 |
| DH2 | Depth Hoar | 1.6 | 364 | 15.9 | 3.67 |
| DH3 | Depth Hoar | 1.6 | 364 | 16.5 | 3.89 |
| RG1 | Rounded Grains | 1.6 | 289 | 23.0 | 3.29 |
| RG2 | Rounded Grains | 1.6 | 304 | 23.7 | 3.81 |
| RG3 | Rounded Grains | 1.6 | 325 | 20.6 | 3.63 |
| RGlr1 | Large Rounded Grains | 1 | 530 | 10.1 | 13.20 |
| RGlr2 | Large Rounded Grains | 1 | 544 | 10.3 | 11.78 |
| RGlr3 | Large Rounded Grains | 1.6 | 557 | 9.9 | 8.99 |
| RGlr4 | Large Rounded Grains | 1 | 542 | 9.3 | 14.01 |
| RGlr5 | Large Rounded Grains | 1 | 541 | 9.7 | 20.49 |
| RGlr6 | Large Rounded Grains | 1 | 526 | 10.1 | 17.22 |

RGlr samples. We estimated the KM coefficients of each sample and averaged them over each snow type. Thus, drift, diffusion functions and jump parameters and their errors are estimated. The errors are reported as the standard error of the means.

### 3.1.2 Results

According to the description provided in Sect. 2, the drift $D^{(1)}(R')$, diffusion $D^{(2)}(R')$, as well as the fourth order KM-coefficients $K^{(4)}(R')$ for normalized data are determined for the four different snow types, PP, DH, RG and RGlr as shown

in Fig. 8. In addition, the autocorrelation function (AFC) is determined from the signals. If the fluctuations of $R'(z)$, belongs to the diffusion processes, one expects that $K^{(4)}(R') = 0$. However, we find that in general, $K^{(4)}(R') \neq 0$, the higher order KM-coefficients are not negligible. It indicates the presence of discontinuities in the snow hardness profile, so that the jump-diffusion model is considered. Therefore, we estimate jump parameters, i.e., jump amplitudes $\sigma_\xi^2(R')$ and jump rates $\lambda(R')$ from data of $R'(z)$ in Fig. 9.

As shown in Fig. 8, the drift coefficients, $D^{(1)}(R', z)$ are mostly linear functions with negative slopes, which describe how fast the system tends back to the stable fixed point. Due to our normalization the fixed point of dynamics is located at the origin, i.e., $R' = 0$. Taking $D^{(1)} = -\gamma R'$, the correlation length scales is given by $L_C = \frac{1}{\gamma}$. For each snow type, $L_C$ is determined for $-2 < R' < 2$: (PP, DH, RG, RGlr) = $(0.01, 0.04, 0.02, 0.08)$ mm. This results agrees with the autocorrelation functions (ACF)
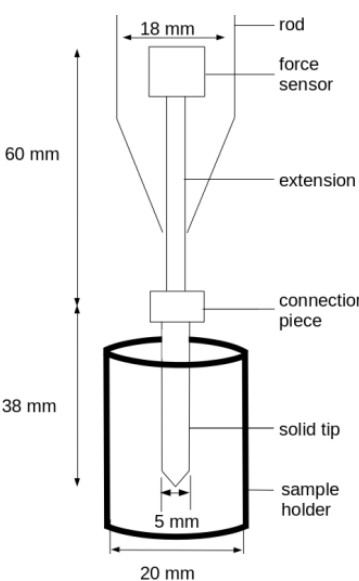

**Figure 5.** Setup of cone penetration test (Peinke et al., 2020). The cone penetration tests (CPT) are conducted by inserting a cylinder of diameter 5 mm, with a conical tip of an apex angle of 60°, into the snow samples. The samples are placed in the cylinder sample holder with a diameter of 20 mm. The cone is inserted vertically at a constant speed of 20 mm s$^{-1}$. The SMP force sensor (Kistler 9207) measures forces in the range of [0, 40] N with a resolution of 0.01 N.

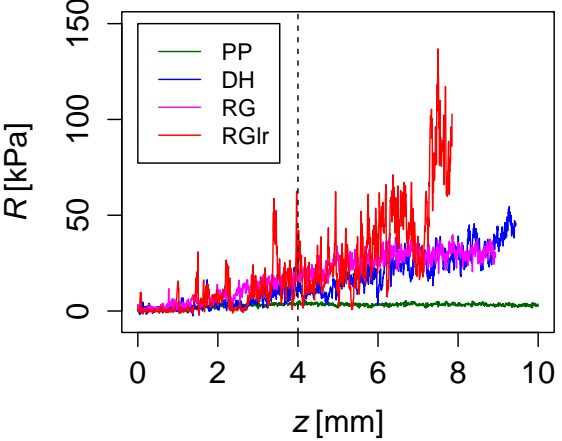

**Figure 6.** Segments of snow hardness profiles of PP1, DH1, RG1 and RGlr1. These four different types occur as natural snow types. Precipitation particles (PP) have smallest trend and fluctuation force, large rounded grains (RGlr) are the largest while depth hoar (DH) and rounded grains (RG) have similar trend and fluctuation size between that of PP and RGlr. The first 4 mm are affected by the progressive penetration of the conic tip and are not considered in the stochastic analysis.





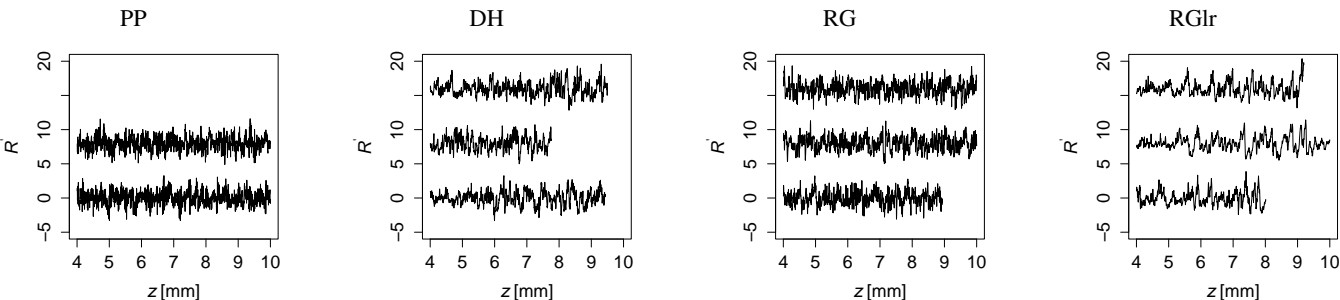

**Figure 7.** Detrended snow hardness profiles of four different snow types, PP, DH, RG and RGlr. The detrended profiles $R'(z)$ of each snow type are shifted vertically for a better visualization.

**Table 2.** Summary of the results for correlation length scales $L_C = \frac{1}{\gamma}$, $L_J = \frac{1}{\lambda}$, jump amplitude $\sigma_\xi^2$ and diffusion and jump ratio $\frac{D^{(2)}}{\lambda\sigma_\xi^2}$ for all snow types analyzed. The results are evaluated in the range of $-2 < R' < 2$.

| Snow type | $L_C = \frac{1}{\gamma}$ | $L_J = \frac{1}{\lambda}$ | $\overline{\sigma_\xi^2}$ | $\overline{\frac{D^{(2)}}{\lambda\sigma_\xi^2}}$ |
|---|---|---|---|---|
| | [mm] | [mm] | | |
| PP | 0.01 | 0.006 | 1.21 | 0.12 |
| DH | 0.04 | 0.01 | 0.54 | 0.48 |
| RG | 0.02 | 0.007 | 0.68 | 0.19 |
| RGlr | 0.08 | 0.02 | 0.38 | 0.53 |

as shown in the same figure. The snow types PP and RG have the shorter correlation length scale, in comparison to the snow types DH and RGlr.

The jump amplitudes, $\sigma_\xi^2(R')$, and the jump probabilities $\lambda(R')\Delta z$, are shown in Fig. 9. The jump amplitude, $\sigma_\xi^2(R')$, indicates how large the jump noise for different $R'$ is. The jump probability describes how probable jumps or discontinuities in forces can occur. To analyze whether diffusion or jump noise is dominating, the dimensionless ratio of diffusion and jump parameters $\frac{D^{(2)}}{\lambda\sigma_\xi^2}$ (Fig. 9 bottom row) is calculated. For a rough estimation, the mean values are determined in the range of $-2 < R' < 2$ and are plotted as blue horizontal lines. The mean of $\bar{\lambda}$ can be used to define second characteristic length scale $L_J = \frac{1}{\lambda}$ (apart from $\frac{1}{\gamma}$). For the above mentioned range of $R'$ we obtain:(PP, DH, RG, RGlr) $= (0.006, 0.01, 0.007, 0.02)\,\text{mm}$. All the results are summarized in Table 2. A discussion of these results will be given in Sect. 4.

## 3.2 Snow Hardness on Field Measurement

### 3.2.1 Measurement Data

Next, measurements from a field campaign are presented (Hagenmuller and Pilloix, 2016). The measurements are performed also with a SMP but the tip had a different sensitivity. The spatial sampling is again 4 µm. This difference in the measurement

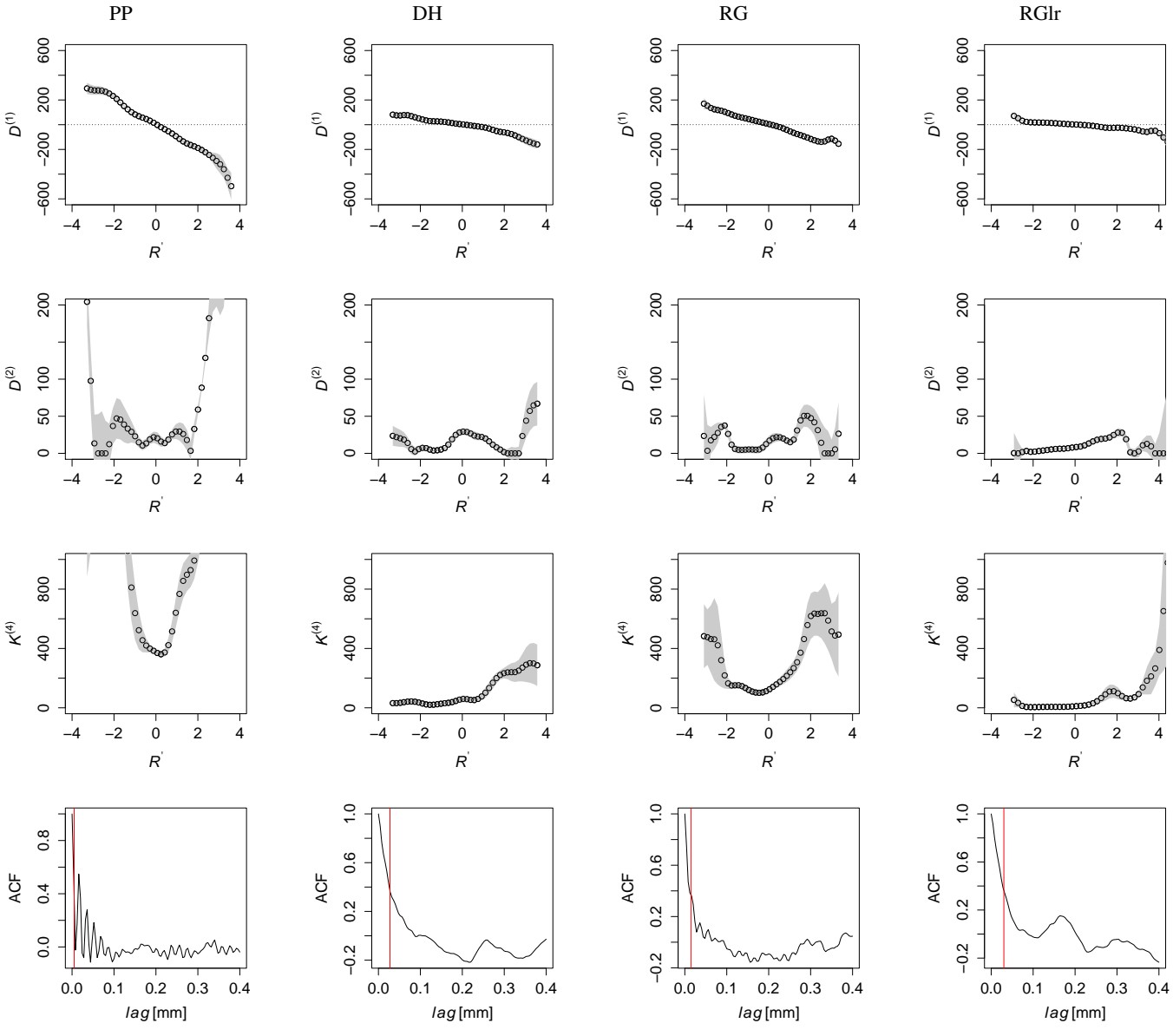

**Figure 8.** State dependent drift $D^{(1)}(R', z)$, diffusion $D^{(2)}(R', z)$, the fourth order KM-coefficients $K^{(4)}(R', z)$, and their respective auto-correlation functions (ACF) of four different snow types, PP, DH, RG and RGlr (left to right). The errors are shown as gray shade background. The red lines in ACF plots indicates the correlation length scales determined from ACF. Note the same ordering of their values is found like for the correlation length estimated from $D^{(1)}$.

methods is of no big importance, as we aim to show with these preliminary results that in principle the stochastic methodology can also be applied to real snow data and that qualitatively comparable results are obtained.

**Figure 9.** Jump amplitude $\sigma_\xi^2(R')$, jump probabilities $\lambda(R')\,\Delta z$ and diffusion and jump ratio $D^{(2)}/\lambda\sigma_\xi^2$ of four different snow types, PP, DH, RG and RGlr (left to right). More data are present in the range of $-2 < R' < 2$, we focus our statistical analysis in this range with less uncertainties. The errors are shown as gray shade background. The diffusion to jump ratios $D^{(2)}/\lambda\sigma_\xi^2$, $D^{(2)}/\lambda\sigma_\xi^2$ for PP and RG are minimum near zero while maximum for DH and RGlr which means that the larger the ice structure, the stronger the diffusion noise, and vice versa. The blue horizontal line shows the correlation length $\frac{1}{\bar\lambda}$, where $\bar\lambda$ is the mean value of $\lambda$ in the range of $-2 < R' < 2$.

### 3.2.2 Results

The snow hardness profile of a field measurement is shown in the left top row of Fig. 10. The measurement profile is strongly inhomogeneous, therefore, we are using the Nadaraya-Watson estimator to determine the local characteristics of the profile. Using the moving-window technique, the profile is separated into non-overlapping window of 500 data point (2 mm) and the detrending is performed on each window by Gaussian kernel with a kernel size of 0.6 mm and normalized with its standard deviation as in previous analysis on laboratory data. For each depth value $z$ and the corresponding value $R'(z)$ the local values
of fourth order KM-coefficient $K^{(4)}(z)$ and the jump amplitude $\sigma_\xi^2(z)$ can be determined, as shown in Fig. 10. The local





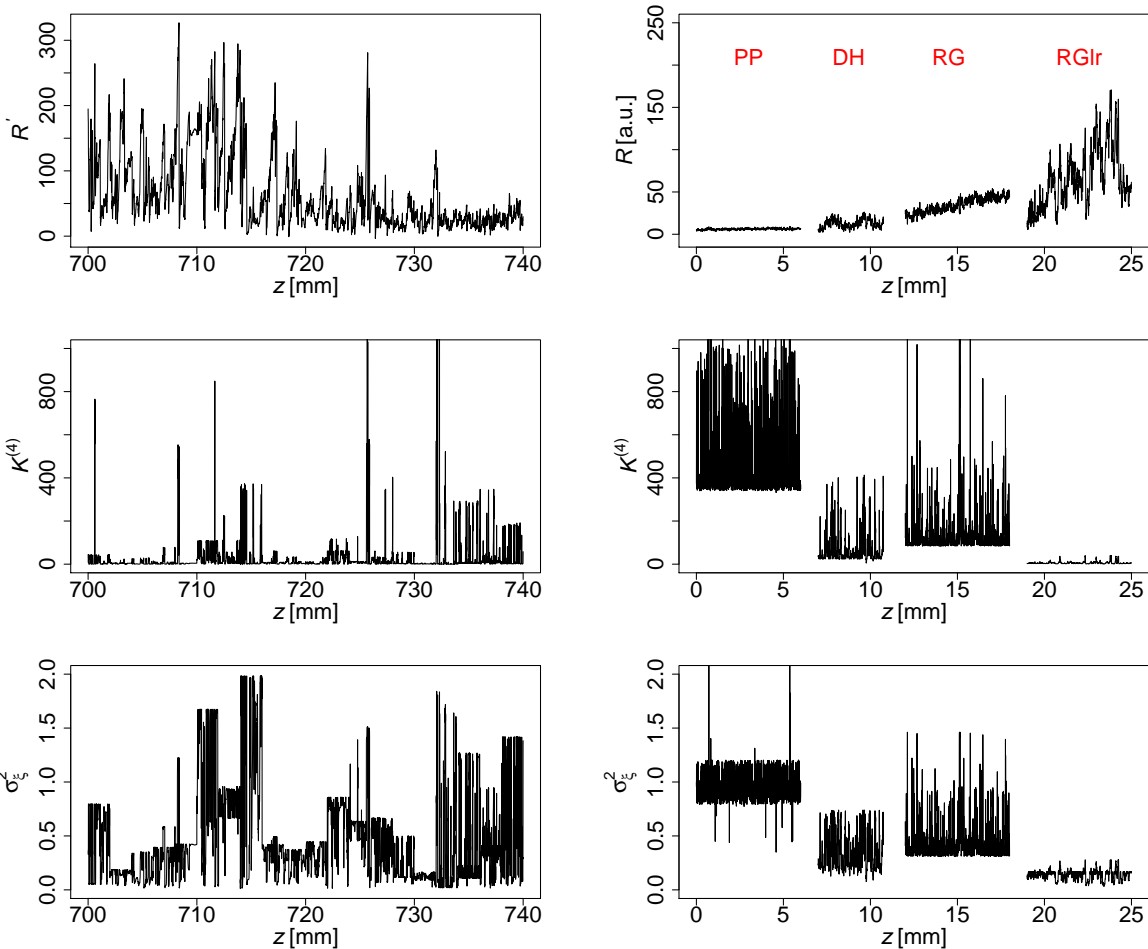

**Figure 10.** The snow hardness profile of a field measurement together with its fourth order KM-coefficients $K^{(4)}$ and the jump amplitude $\sigma_\xi^2(z)$ are shown in the left column. These parameters are determined using non-overlapping moving window with 500 data points (2 mm) by means of Nadaraya-Watson estimator. Snow hardness profiles of the laboratory prepared snow types and their local parameters are also plotted to have a better comparison (right column). They are shifted horizontally for better visualization. The results of field measurements are shown at the depth of $700\,\text{mm} < z < 740\,\text{mm}$. With reference to the local characteristic of different snow types from the laboratory measurements, we can see the dynamics that mixtures of different snow types are present in this section of measurement. For $732\,\text{mm} < z < 740\,\text{mm}$, the high $K^{(4)}$ and $\sigma_\xi^2$ indicate the presence of small and less dense structures of snow which resembles the RG-like snow types.

characteristic of each snow type from the previous section is plotted in the right column of Fig. 10 for a better comparison with the field measurement data. The interpretation of the results will be discussed in the next section.





## 4 Discussion

Our work is based on the proposed analogy of Brownian motion and the SMP penetration process, as illustrated in Fig. 1.
The events of bond-breaking or, respectively, of collision with molecules sum up in a mean force and noise. For continuous Brownian noise, we need an integration over sufficient micro-scale events, as discussed by Einstein (1905) in his original paper. Commonly it is found that sufficient large particle leads to this integration, see Fig. 1(b). To our interpretation, this integration over discrete single events of bond-breaking in the surrounding of SMP and, in addition, pushing aside of loose snow grains during the penetration process form continuous Brownian noise. However, the jump noise should be more appropriate for the
bond-breaking events directly at the tip of SMP and the amplitude of the jump noise should depend on the strength of the ice bonds and void sizes. From this interpretation, it is clear that the morphology of the snow types, shown in Fig. 4, is essential for the stochastic analysis outlined in this paper.

We start the discussion with the mean values of $R$ and the standard deviation $\sigma_R$ (see Fig. 6 and Table 1). The less-dense PP and the dense RGlr snow can be well separated, whereas the differences are less clear for DH and RG. In the following, we
discuss the measured SMP penetration profiles based on our stochastic results. Since we now focus on a stochastic investigation of the fluctuations of the penetration profiles, the detrended and normalized data are $R'$ used. Furthermore, we can note that the normalization of the snow profiles does neither affect the correlation length scales $L_C = \frac{1}{\gamma}$ from the drift coefficients nor the jump characteristic length scales $L_J = \frac{1}{\lambda}$. Because this analysis depends on the amount of available data, our discussion of the estimated KM coefficients is limited to the range $-2 < R' < 2$.

The drift terms $D^{(1)}$, Fig. 8, are all monotonously decaying with increasing $R'$ and as already mentioned, can be approximated by a linear decay, $D^{(1)} = -\gamma R'$. The slope indicates how fast the signals relax to fixed point located at $R' = 0$. The magnitudes of the slopes are PP > RG > DH > RGlr, thus the PP-snow has fastest relaxation or shorter correlation length scale $L_C$. We find that the larger the ice structures, the slower (or longer) the relaxation. If we compare this result with the snow structures shown in Fig. 4, we conclude that $L_C$ or, respectively, $\gamma$ is clearly related to the size of the snow structures.

The results for $D^{(2)}$ show that about the same diffusive noise amplitude is found for all snow types. In contrast, we see clear differences for the fourth order KM-coefficients $K^{(4)}$. Although always $K^{(4)} \neq 0$, clear differences in the magnitude of this KM-coefficient are found. $K^{(4)}$ is the largest in PP, followed by RG, DH and RGlr, respectively.

The amplitudes of jump noise, $\sigma_\xi^2$, is the highest for PP, followed by RG, DH and RGlr, respectively. For the jump probabilities $\lambda \Delta z$, we distinguish a group composed of PP and RG, with higher values, and one composed of DH and RGlr, with lower
values. One can interpret this finding such that for the precipitation particles (PP, recent snow) with very small ice structures and high porosity, the breaking occurs easily and frequently which explains that the jump probability has the largest contribution here. Similarly, RG is also less dense with smaller ice structures than DH and RGlr. Thus, we can also interpret this finding such that the smaller the ice structures and the less densely packed snow, the stronger the jump noise. For the densely packed snow with larger ice structures, the breaking of the ice structures is less frequent, which explains a lower jump probability. The
mean of jump rate $\bar{\lambda}$ in the range of $-2 < R' < 2$ can be used to define second characteristic length scale $L_J = \frac{1}{\lambda}$. Similarly to the correlation length scale $L_C = \frac{1}{\gamma}$, PP has the smallest length, followed by RG, DH and RGlr.





Besides the features of the different terms in the stochastic processes, the contributions of the diffusive and the jump noise can be compared by the dimensionless quotient of $\frac{D^{(2)}}{\lambda \sigma_\xi^2}$, i.e., the relation between the two noise contributions. Consistent with our discussion above, the smallest values for $\frac{D^{(2)}}{\lambda \sigma_\xi^2}$ are obtained for the PP, i.e., the jump noise is dominating due to the frequent

fracture of small (soft) ice structures. For the other snow types, we see that within the range of $-2 < R' < 2$ the values of $\frac{D^{(2)}}{\lambda \sigma_\xi^2}$ increase with larger ice structures, in accordance with Fig. 4. The quotient $\frac{D^{(2)}}{\lambda \sigma_\xi^2}$ is smaller for PP and RG while larger for DH and RGlr. For RGlr, the diffusive noise dominates in a broad range of $R'$- values. For the densely packed snow with bigger ice structures, it also takes much force to push the ice grains on the side but not necessarily to break the cohesive bonds close to the tip. Therefore the penetration signal is dominated by Brownian noise. This result is also consistent with the value of $K^{(4)}$

which is relatively smaller for RGlr. It is interesting to see that $\frac{D^{(2)}}{\lambda \sigma_\xi^2}$ and $K^{(4)}$ allows to differentiate RG and DH. In constrast, according to the classical statistical features of the snow signals shown in Tab. 1, the differences for DH and RG are less clear.

In Section 3.2, we analyse field measurement data which is highly inhomogeneous. With reference to the local characteristic of different snow types from the laboratory measurements (right column of Fig. 10), we can see the dynamics that mixtures of different snow types are present in this section of measurement. For $732\,\mathrm{mm} < z < 740\,\mathrm{mm}$, the high $K^{(4)}$ and $\sigma_\xi^2$ indicate

the presence of small and less dense structures of snow which resembles the RG-like snow types. Based on these preliminary results of real field data the developed methodology thus appears promising to interpret cone penetration tests in the filed but quantitative evaluation remains to be done.

## 5   Conclusions

In conclusion, we observe that the advanced stochastic analysis of SMP measurements of snow layers allows differentiating

snow types. The diffusive and jump-noise contribution can be quantified and give new insights into the stochastic behaviors of the cone penetration test in snow. For different snow types, we find an interesting mixture of diffusive- and jump-like noise. We propose the interpretation that the dominant diffusive process is due to the pushing aside of many snow grains, whereas the breaking of ice structures leads to dominant jump noise. Our results show that the denser structures like for DH and RGlr lead to a more collective diffusive behavior, whereas for the highly-porous snow structures of PP and RG, the single breaking

events lead to a relatively stronger jump noise. For this interpretation, we have to remember that all $R$-values are detrended and normalized, thus the absolute values of the snow hardness $R$ are not essential but more the resulting collective behavior of the snow types.

As a last comment, we would like to point out that our characterization of a complex matter, here snow, by a penetration process should have the potential to be generalized to, for example, biological tissue or ground layers. Last but not least, we

would like to point out that our work provides additional insight into analyzing and modeling the complex nature of snow types, but our intention is not to replace existing methods.

*Code and data availability.*  The code and data for this analysis can be found under https://github.com/pplin13/Snow_RCodes





*Author contributions.* PPL did the preliminary analysis of the data and simulations, and wrote the main part of the manuscript. PH and IP performed the experiments. JP had the initial idea, and JP, MW and MRRT supervised the work. All the authors interpreted the results, and
helped in preparing and editing the manuscript.

*Competing interests.* The authors declare no conflict of interest.

*Acknowledgements.* The authors would like to thank Christian Behnken and Hauke Hähne for the helpful discussions.





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
