# Peer review of "Stochastic analysis of micro-cone penetration tests in snow"

_The Cryosphere, 2022_

## Referee Comment (RC2)

**Review of "Stochastic analysis of cone penetration tests in snow" by Lin et al**

**Main comments**

The paper analyzes snow micro-penetrometer (SMP) signals in terms of stochastic jump-diffusion dynamics. The analysis is very interesting and adds novel aspects to the interpretation of SMP data. The paper is well written, the topic is suitable for TC and the results warrant publication.

I have mainly two questions I would like to ask an answer for in the present work:

- **Independence**. Naively one is tempted to assume that the signal must be the result of one (and only one) underlying stochastic process, which is the disordered microstructure of snow. The present model rather assumes that the penetration leads to a situation with contributions from different stochastic processes which are assumed to be independent. While I can imagine how such a situation may originate from the physics, it would be helpful if this assumption of independence (between the jump characteristics and the diffusion) could be further assessed.

  Along these lines we have previously seen that interpreting the signal as a shot noise process with three parameters ($\lambda$, $\delta$ and $f_0$) we always end up with correlations between estimates of $\lambda$, and $\delta$, which obviously cannot be convincingly separated by such a model. Since $\delta$ roughly translates to $1/D^{(1)}$ of the present model, I wonder if these parameters still show correlations. In addition, for large $\lambda$ and small $\sigma_\xi$ a jump process with drift could "tend" to diffusion. Therefore potential correlations of the latter parameters with $D^{(2)}$ are relevant too. In the simplest case it would be sufficient to provide mutual scatter-plots of estimated parameters $\lambda$, $D^{(1)}$, $\sigma_\xi$ and $D^{(2)}$ for the profile in Sec 3, but maybe there are even rigorous ways to answer this question.

- **Kernel width**. It might be good to check if a fixed kernel width of 0.6mm (l168) is a robust choice in view of the statements in the discussion about grain size dependencies: Converting the SSA values given in Tab 1 into a "grain size" (the optical diameter) reveals that diameters range from 0.12mm for PP to 0.70 mm for RGlr. Now the diffusive contribution is assumed to be a result of steric (grain-grain) interactions in front of the cone, and this process will need a few grain diameters to develop. A fixed kernel of this particular size might thus induce a bias here. *A priori* grain size information is clearly commonly not available (like for the analysis of hardness profile in Sec 3.2). But it seems relevant to compare, at least for the data from SEc 3.1, how the parameter estimates for the 4 samples compare with those generated from a *constant* ratio of kernel width and optical diameter. Results could be simply added to existing figures.

Kind regards,
Henning Löwe

**Minor comments**

(l117): I don't entirely understand why an $R$ dependence of the coefficients is introduced here. Isn't the analysis later only based on constant coefficients, i.e. additive noise? Would everything work also for multiplicative noise?

(l135): Here it might be illustrative to explicitly mention the "triply-stochastic" nature of Eq 11 and that all $(\xi, J_t, W_t)$ are independent.

(Tab 2): Here uncertainties/errors should be included that reflect inter-sample variations of the same snow type.

(l174): What is the final size of the sub-samples? Is this choice also consistent with grain size $\ll$ sample size in all cases?

(l189): It would be nice to include the correlation lengths estimated from the ACF also in Tab 2 to support this statement. (DH and RGlr appear to be very similar in Fig 8 while the $L_c$ differ by a factor of two)

(Fig 9): What is taken as $\Delta z$?

(Fig 10): Top left, this looks like $R$ and not $R'$?

(Fig 10): Maybe a semilogy scale for $K^4$ better reveals the differences?

(Fig 10): It would be good to include also $\lambda \Delta z$ and $D^{(2)}/\lambda \sigma_\xi^2$ in this figure. The subfigures can be safely reduced a bit in height.

(l234): This is such a statement which might be affected by the choice of the kernel width...

---

## Author Comment (AC1)

**Reply to Referee Comments 1**

*We first would like to thank the reviewer (Adrian McCallum) for the positive and insightful as well as detailed comments. We take all the comments into account, reply in italic text and will update accordingly in the revised manuscript.*

**General comments**

Thank you for the opportunity to review this very interesting work that examines how variations in stochastic signals resulting from micro cone penetration in snow can be used to discern snow type/microstructure.
Field data is compared with laboratory data to enable characteristic signals to be identified and variation in two noise types (diffusive and jump) is particularly examined to suggest snow microstructure behaviour and thus snow type/composition. This work relies on the assumption that the SMP penetration process is analogous to Brownian motion.
I found it a very interesting paper and I recommend it for publication. Below I make a few specific comments and numerous technical observations which the authors may wish to consider.

**Specific comments**

My primary comment is that you may wish to consider altering the title/context/frame of your paper. I say this because McCallum has written many papers on Cone Penetration Testing (CPT) in polar snow, some of which examine microstructure assessment using CPT. You may wish to briefly comment on these works in your introduction, or you may wish to refer to your work as 'micro' cone penetration testing, just to differentiate between the large body of McCallum's large-scale (36.7 mm penetrometer) work and the body of work that you discuss here, primarily pertaining to the SMP, Johnson and Schneebeli's work etc. I am happy with whatever you chose to do; but, if you keep it as cone penetration tests, you probably should mention McCallum's work...

*The title is now changed to:*
*"Stochastic analysis of micro-cone penetration tests in snow"*
*and McCallum's works is also mentioned in the introduction.*

The rest of my comments are essentially of a technical nature.

**Technical observations**

Please re-examine your tense throughout the document. You start off in past tense but this alters; please review and amend.

*We check the used tense accordingly in the revised manuscript.*

Now by line #, for your consideration please:

"more and more" etc. please re-phase/tighten this sentence.

*We now change it to "By using small penetrometer tips at this high vertical resolution, further details of the penetration process get resolved, leading to much more stochastic signals."*

delete "employing"

*Corrected.*

replace allows with enables

*Corrected.*

probably less-dense not lighter; keep terminology consistent.

*We now use "less-dense"*

single: how do you discern/confirm this? Perhaps reword.

*We now outline that these are our interpretation and findings of our analysis.*

Perhaps: with micro cone penetration tests.

*We specifically mention now as "micro-cone penetration tests"*

Perhaps: supposedly

*Corrected.*

can be resolved

*Corrected.*

Reference re. important applications?

*The text is reworded as "These models are now commonly used to characterize the snowpack stratigraphy from SMP measurements."*

to some of the most...

*Corrected.*

delete "the" in the fluid
*Corrected.*

what are micro-events? Please better explain.
*We reformulate the sentence and explain it in the revised manuscript. "Due to the sum of several collisions with the molecules in the fluid as illustrated in Fig. 1, the large red particle undergoes a motion described by a stochastic process."*

shot noise? correct?
*Our interpretation of Poisson jump noise corresponds to the shot noise. We mention it now in the revised manuscript.*

Please reword this last sentence; perhaps: Via this advanced analysis, we seek more detailed snow characterisation from micro cone penetration test resistance data.
*Done. We used your suggestion*

Delete "the" Sect. 3
*Corrected.*

explains the equations?
*Corrected.*

64/5 note that although the drive is constant the actual penetration rate may not be.
*We agree. In the paper, we do not mention the penetration rate.*

fix "as of the"
*Corrected.*

probably just Friedrich () and Rinn () (instead of semi-colon).
*Corrected.*

do you mean: small depth interval (z)? Also "similar"
*Yes and corrected.*

probably: Such a jump-diffusion dynamic. . .

*Corrected.*

Wick's theorem: reference?
    *Added.*

where here we use. . .
    *Corrected..*

do you mean: small depth interval (z)?
    *Yes and corrected.*

perhaps state: "; this is the same as Eq. 2 but. . ."
    *Corrected..*

perhaps "is considered instantaneous".
    *Corrected.*

Please spell out OU and SDE in Fig. 2 caption.
    *Done.*

Please use drift-jump and jump consistently so as not to cause confusion.
    *We now change jump to drift-jump.*

Rephrase "as above"; this is unclear.
    *Corrected. "as above" is now deleted.*

"which is a zero-one. . ."
    *Corrected.*

137/138 etc. "process were generated"; please change tense to past throughout.
    *Corrected and we now change the tense throughout the paper accordingly.*

139/140 "Left, a pure. . . , middle, . . . and right, . . . "
    *Corrected.*

negligibly
    *Corrected.*

"Dots" in Fig. 3 caption
*Corrected.*

process, another parameter that we considered was...
*Corrected.*

proof evidence? Perhaps: to validate our method, based on the KM coefficient...; then. comma after "Eq. 8"; "were chosen"
*Corrected.*

"as the previous example"
*Corrected.*

Probably: Firstly, small snow samples whose microstructure was fully characterised... were used to test... Secondly,... we analysed one... and provided...
*Corrected*

Fig. 4 caption; final sentence: Sub-samples shown are ...
*Corrected.*

tested
*Corrected.*

Reference for snow types; the samples were prepared.
*The reference Fierz et al. (2009) is added.*

Temperature of sintering? Microstructure was captured.
*Sintering temperature is -10 °C and added in the revised manuscript.*

test was conducted
*Corrected.*

on sample preparation..
*Corrected.*

Main sample properties are summarised in Table 1 and the measured hardness profiles...
*Corrected.*

focussed on the fluctuations of the hardness profiles. Each profile was first detrended.

    *Corrected.*

divided

    *Corrected.*

were separated

    *Corrected.*

We estimated the KM coefficients of each sample...; how?

    *We use Eq. 2 to estimate them and mention it in the revised manuscript.*

data were determined

    *Corrected.*

was determined

    *Corrected.*

"... 0, and the higher order KM..."

    *Corrected.*

This indicates the presence...

    *Corrected.*

normalizaton, the fixed...

    *Corrected.*

length scale is given

    *Corrected.*

Figure 5 caption: Setup of micro cone penetration test; The samples were placed in the cylindrical sample holder...; Is "Kistler 9207" the type of force sensor?

    *Corrected in the figure caption. Yes, "Kistler 9207" is the force sensor used in SMP.*

Figure 6 caption. The wording here is unclear: "have smallest trend and fluctuation force"; are you using all these terms consistently? In the next sentence you talk of size not force? Please re-examine...

*We now use fluctuation "force" in the figure caption 6.*

Figure 7 caption. . . . for better visualisation.
   *Corrected.*

Perhaps: Results are summarized in Table 2; we discuss these in Sect. 4.
   *Corrected.*

Perhaps: Hardness of Field Data or Application to Field Snow Data?
   *we change the title of 3.2 to "Application to Field Snow Data".*

The measurements were also performed with a SMP, but the tip had a different sensitivity of. . . what was it?? Spatial sampling was again. . .
   *we reword the text as "the tip had a slightly different shape corresponding to the standard version of the SMP (Johnson & Schneebeli, 1999)."*

Figure 8 caption. Please reword last sentence; it is difficult to understand.
   *we now change it to "Comparing the correlation length scales $L_{\mathrm{C}} = \frac{1}{\gamma}$ where $D^{(1)} = -\gamma R'$ with those of the autocorrelation functions (ACF), we find that both length scales have the same ordering of their values for all snow types."*

methods was irrelevant, as we subsequently show. . . that in principle, the. . . really snow data, and that. . .
   *Corrected.*

Figure 9 caption. "$< 2$; we focus our statistical. . . "
   *Corrected. The last line of the caption is also changed to "The blue horizontal lines show the mean values of the respective parameters in the range of $-2 < R' < 2$."*

therefore, we used. . .
   *Corrected.*

profile was separated. . . and detrending was performed on each window. . . 0.6 mm, formalised with. . . deviation as in our previous analysis of laboratory data.

*Corrected.*

Figure 10 caption. parameters were determined... are also plotted to enable better comparison (right column); they are shifted... reference to the local characteristic snow types from laboratory measurements,
 *Corrected.*

for better comparison
 *Corrected.*

Interpretation of these results will be discussed next.
 *Corrected.*

"it is found that sufficient large particle"?? Please reword. "In our interpretation,..."
 *Corrected.*

perhaps: "in the immediate surroundings of the SMP, in addition to the pushing aside..."
 *Corrected.*

Delete However; Perhaps: The jump noise may represent (or be representative of) the bond-breaking events occurring directly at the tip of the SMP...
 *Corrected.*

perhaps: it is clear that snow type morphology, shown in Fig. 4, is essential for effective stochastic analysis as outlined herein.
 *Corrected.*

We started...
 *Corrected.*

"$R'$, and can be approximated..."
 *Corrected.*

our earlier discussion,
 *Corrected.*

"bigger ice structures": consider rewording/clarifying this sentence: "thicker grain necks"?

*We now use "larger grain size".*

"allows"? perhaps: enables differentiation between...

*Corrected.*

perhaps: "With reference to the local characteristic snow types from the laboratory measurements (), we see dynamics that suggest mixtures of different snow types within this depth segment.

*Corrected.*

"the developed methodology appears... in the field, but further quantitative evaluation is required."

*Corrected.*

allows differentiation of

*Corrected.*

the denser structures typical of DH and...

*Corrected.*

Delete: "we have to remember that"

*Corrected.*

Perhaps: Finally, we would... of a complex material, snow, by a...

*Corrected.*

Perhaps: types, complementing existing methods.

*Corrected.*

---

## Author Comment (AC2)

**Reply to Referee Comments 2**

*We first would like to thank the reviewer (Henning Löwe) for the positive and insightful comments. We take all the comments into account, reply in italic text and will update accordingly in the revised manuscript.*

**Review of "Stochastic analysis of cone penetration tests in snow" by Lin et al**

**Main comments**

The paper analyzes snow micro-penetrometer (SMP) signals in terms of stochastic jump-diffusion dynamics. The analysis is very interesting and adds novel aspects to the interpreation of SMP data. The paper is well written, the topic is suitable for TC and the results warrant publication. I have mainly two questions I would like to ask an answer for in the present work:

- Independence. Naively one is tempted to assume that the signal must be the result of one (and only one) underlying stochastic process, which is the disordered microstructure of snow. The present model rather assumes that the penetration leads to a situation with contributions from different stochastic processes which are assumed to be independent. While I can imagine how such a situation may originate from the physics, it would be helpful if this assumption of independence (between the jump characteristics and the diffusion) could be further assessed. Along these lines we have previously seen that interpreting the signal as a shot noise process with three parameters ($\lambda$, $\delta$ and $f_0$) we always end up with correlations between estimates of $\lambda$, and $\delta$, which obviously cannot be convincingly separated by such a model. Since $\delta$ roughly translates to $1/D^{(1)}$ of the present model, I wonder if these parameters still show correlations. In addition, for large $\lambda$ and small $\sigma_\xi$ a jump process with drift could "tend" to diffusion. Therefore potential correlations of the latter parameters with $D^{(2)}$ are relevant too. In the simplest case it would be sufficient to provide mutual scatter-plots of estimated parameters $\lambda$, $D^{(1)}$, $\sigma_\xi$ and $D^{(2)}$ for the profile in Sec 3, but maybe there are even rigorous ways to answer this question.

  *Thank you for the interesting question. We look at the scatter-plots of the parameters $\frac{1}{\gamma}$ and $\lambda$, and $\frac{1}{\gamma}$, $D^{(1)}$, $\sigma_\xi^2$ and $D^{(2)}$ for all four snow types according to the results of Table 2. We also do the linear*

*regression of each scatter-plots and the coefficient of determination or R-squared is ≈ 0.9. In our current analysis, we assume the independence between diffusion and jump-diffusion which could give a good insight to interpret the cone penetration process. However, from these plots, we could see the hints for the possible correlations which could be considered as the improvement for current model and we would leave it as an open topic for the later analysis.*

*In the jump-diffusion modeling of stochastic time series it is assumed that three random variables, $W(t)$ Wiener process, $\xi$ jump size and $J(t)$ Poisson jump process, are independent. However in general they can be correlated. For instance for correlated $W(t)$ and $J(t)$ one finds*

$$\langle W(t)J(t) \rangle = \rho(t)\sqrt{\lambda}t$$

*where $\rho(t)$ is the correlation coefficients of $W(t)$ and $J(t)$.*

*Data-based estimation of $\rho(t)$ is an open topic and we will address this important problem in the near future.*

[Figure]

[Figure]

[Figure]

[Figure]

- Kernel width. It might be good to check if a fixed kernel width of 0.6 mm (l168) is a robust choice in view of the statements in the discussion about grain size dependencies: Converting the SSA values given in Tab 1 into a "grain size" (the optical diameter) reveals that diameters range from 0.12 mm for PP to 0.70 mm for RGlr. Now the diffusive contribution is assumed to be a result of steric (grain-grain) interactions in front of the cone, and this process will need a few grain diameters to develop. A fixed kernel of this particular size might thus induce a bias here. A priori grain size information is clearly commonly not available (like for the analysis of hardness profile in Sec 3.2). But it seems relevant to compare, at least for the data from Sec 3.1, how the parameter estimates for the 4 samples compare with those generated from a constant ratio of kernel width and optical diameter. Results could be simply added to existing figures.

Here we calculate the grain size or optical diameter based on the relation $d_{\text{opt}} = \frac{6}{\rho_{\text{ice}} \cdot \text{SSA}}$. Then, we calculate the average value of SSA for each snow types and find the average grain sizes of {PP, DH, RG, RGlr} to be {0.14, 0.40, 0.29, 0.66} mm. Now we use these grain sizes as the kernel widths for detrending of the snow hardness profiles and the new results for Fig 8 and 9 and Table 2 are as follow:

[Figure]

[Figure]

| Snow type | $L_{\mathrm{C}} = \frac{1}{\gamma}$ | $L_{\mathrm{J}} = \frac{1}{\lambda}$ | $\overline{D^{(2)}}$ | $\overline{\sigma_\xi^2}$ | $\overline{\frac{D^{(2)}}{\lambda\sigma_\xi^2}}$ |
|---|---|---|---|---|---|
| | [mm] | [mm] | | | |
| PP | 0.008 | 0.005 | 28.59 | 1.40 | 0.12 |
| DH | 0.035 | 0.010 | 15.66 | 0.59 | 0.44 |
| RG | 0.017 | 0.006 | 15.94 | 0.73 | 0.14 |
| RGlr | 0.080 | 0.016 | 10.24 | 0.36 | 0.53 |

*The new kernel widths do not change the results significantly. Therefore, we decide to keep the constant kernel width of 0.6 mm in order to make it consistent with the field measurement data. The chosen kernel width is also within the range of the smallest and the largest grain size. If we choose the kernel width that is much larger than grain size, there would lead to oversmoothing and the detailed dynamics of cone penetration test could be lost. We also added the remark to the paper, that the results do not change significantly if the kernel width are changed between 0.14 mm and 0.66 mm.*

Kind regards, Henning Löwe

**Minor comments**

(l117): I don't entirely understand why an $R$ dependence of the coefficients is introduced here. Isn't the analysis later only based on constant coefficients, i.e. additive noise? Would everything work also for multiplicative noise?

*Our analysis works for multiplicative noise. From our results we can see that the parameters are $R$ dependent (Fig. 8 and 9). When we interpret the results in Table 2, we took the average value as a first order approximation of the parameters in the range of $-2 < R' < 2$ in order to compare the results of each snow type.*

(l135): Here it might be illustrative to explicitly mention the "triply-stochastic" nature of Eq 11 and that all $(\xi, J_t, W_t)$ are independent.

*We now write: "Here, we have triply stochastic processes $W_t$, $J_t$ and $\xi$ which are all independent of each other."*

(Tab 2): Here uncertainties/errors should be included that reflect inter-sample variations of the same snow type.

*Uncertainties are now included in Table 2.*

(l174): What is the final size of the sub-samples? Is this choice also consistent with grain size $\ll$ sample size in all cases?

*The final sizes of the sub-samples vary from (680 - 1500) sample points i.e. (2.72 - 6) mm. The grain sizes varies approximately from (0.14 - 0.66) mm.*

(l189): It would be nice to include the correlation lengths estimated from the ACF also in Tab 2 to support this statement. (DH and RGlr appear to be very similar in Fig 8 while the $L_c$ differ by a factor of two)(Fig 9): What is taken as $\Delta z$?

*The correlation lengths $L_{\mathrm{ACF}}$ estimated from the ACF, $(PP, DH, RG, RGlr) = (0.006, 0.025, 0.016, 0.038)$ mm, are added in the text and Table 2 in the revised manuscript. $\Delta z$ is the resolution of SMP which is 4 $\mu$m.*

(Fig 10): Top left, this looks like $R$ and not $R'$?

*Corrected.*

(Fig 10): Maybe a semilog $y$ scale for $K^{(4)}$ better reveals the differences?

*Here is the figure for $K^{(4)}$ in semilog $y$ scale and we decide to keep it in linear scale since we can observe the differences in $K^{(4)}$ between the snow*

*types more clearly in linear scale.*

[Figure]

(Fig 10): It would be good to include also $\lambda \, \Delta z$ and $D^{(2)}/\lambda \sigma_\xi^2$ in this figure. The subfigures can be safely reduced a bit in height.

*Since the uncertainties of $\lambda$ are relatively large especially for the extreme values of $R'$, we mainly determine $K^{(4)}$ and $\sigma_\xi^2$ which could give more consistent results.*

(l234): This is such a statement which might be affected by the choice of the kernel width...

*As shown in the main comment, our choice of kernel width does not significantly change the results.*

---

## Author Response (AR2)

*We first would like to thank the editor (Melody Sandells) for accepting our paper for publication in The Cryosphere as well as the positive comments. We also would like to thank the reviewers again for their decision and the time they took to review our paper. We update the suggested technical corrections accordingly in the final manuscript.*

**Technical corrections**

NB line numbers refer to tracked changes version

lines 4 and 26 'much more stochastic signals' -> 'many more stochastic signals' or 'signals of a much more stochasic nature'
   *We now change both of them to "many more stochastic signals"*

line 12 'breaking single cohesive' -> 'breaking of a single cohesive'
   *corrected*

line 30 'penetration test' -> 'penetration tests'
   *corrected*

line 123 'in term' -> 'in terms'
   *corrected*

Line 142-143 remove brackets around sentence.
   *corrected*

Table 1 caption 'images Peinke' -> 'images from Peinke'
   *corrected*

Figure 7. Please add specific RGlr sample numbers to caption (3 shown, table 1 has 6)
   *We add an additional line in the figure caption mentioning which three samples of RGlr we show in the figure. "For the snow type RGlr, we only show the samples, RGlr4, RGlr5 and RGlr6, as described in Tab. 1."*

   *Additionally, we also do some small corrections on the caption of Fig. 9. "Jump amplitudes $\sigma_\xi^2(R')$, jump probabilities $\lambda(R')$ $\Delta z$ and diffusion and jump ratios $\frac{D^{(2)}}{\lambda \sigma_\xi^2}$ of four different snow types ... The diffusion to jump ratios $\frac{D^{(2)}}{\lambda \sigma_\xi^2}$ for PP and RG are minimum ..."*